# A Deep Convolutional Neural Network for the Early Detection of Heart Disease

**DOI:** 10.3390/biomedicines10112796

**Published:** 2022-11-03

**Authors:** Sadia Arooj, Saif ur Rehman, Azhar Imran, Abdullah Almuhaimeed, A. Khuzaim Alzahrani, Abdulkareem Alzahrani

**Affiliations:** 1University Institute of Information Technology, PMAS-Arid Agriculture University, Rawalpindi 46000, Pakistan; 2Department of Creative Technologies, Faculty of Computing & Artificial Intelligence, Air University, Islamabad 42000, Pakistan; 3The National Centre for Genomics Technologies and Bioinformatics, King Abdulaziz City for Science and Technology, Riyadh 11442, Saudi Arabia; 4Faculty of Applied Medical Sciences, Northern Border University, Arar 91431, Saudi Arabia; 5Faculty of Computer Science and Information Technology, Al Baha University, Al Baha 65779, Saudi Arabia

**Keywords:** image classification, deep learning approach, deep convolutional neural network, computer vision, heart disease

## Abstract

Heart disease is one of the key contributors to human death. Each year, several people die due to this disease. According to the WHO, 17.9 million people die each year due to heart disease. With the various technologies and techniques developed for heart-disease detection, the use of image classification can further improve the results. Image classification is a significant matter of concern in modern times. It is one of the most basic jobs in pattern identification and computer vision, and refers to assigning one or more labels to images. Pattern identification from images has become easier by using machine learning, and deep learning has rendered it more precise than traditional image classification methods. This study aims to use a deep-learning approach using image classification for heart-disease detection. A deep convolutional neural network (DCNN) is currently the most popular classification technique for image recognition. The proposed model is evaluated on the public UCI heart-disease dataset comprising 1050 patients and 14 attributes. By gathering a set of directly obtainable features from the heart-disease dataset, we considered this feature vector to be input for a DCNN to discriminate whether an instance belongs to a healthy or cardiac disease class. To assess the performance of the proposed method, different performance metrics, namely, accuracy, precision, recall, and the F1 measure, were employed, and our model achieved validation accuracy of 91.7%. The experimental results indicate the effectiveness of the proposed approach in a real-world environment.

## 1. Introduction

The invention of artificial intelligence was a revolutionary breakthrough for humanity that opened the gateway to a different world [1]. From basic chatbots [2] to autonomous vehicles [3] and robots [4], it has achieved wonders in every domain of life. Artificial Intelligence (AI) has strengthened the complex decision-making process and supports all computer-aided learning. AI is a combination of multiple disciplines, such as logistics, biology, linguistics, computer science, mathematics, engineering, and psychology. It has achieved extraordinary results in the field of speech and facial recognition, natural language processing, intelligent robots, and image recognition [5]. The vision of the human brain and AI was directed to the invention of great machines. These machines have rendered daily human life easier and more convenient. Machine learning is one of the techniques that have rendered this possible. Machine learning is the ability of computers to learn without being explicitly programmed. In this AI technique, computers learn from previous experiences and data [6]. The amount of data is increasing rapidly, so there is a need to efficiently handle the data. Sometimes, it becomes quite difficult for humans to manually extract useful information from raw data due to their inconsistency, uncertainty, imprecision, and alikeness [7]. This is where machine learning is useful. With the profusion of data in the form of big data, its demand is on the rise, as it obtains more accurate, informative, and consistent information from raw data. The main objective of machine learning is to enable machines to learn without being thoroughly programmed [8]. Machine learning has remarkably advanced in many fields, such as preprocessing techniques and learning algorithms, during the last few decades.

Deep learning is one of those remarkable advancements [9] that has rendered AI even smarter. Deep learning is the branch of machine learning that was named in 2006. It was inspired by the structure of the human brain, which contains neural networks. It is a data-processing method that uses a multiple-layer technique [10]. The working of the layers can be considered to be a layer receiving weighted input, transforming it into mostly nonlinear functions, and then sending the output to the next layer [11]. It is aiding in addressing issues that have limited the finest efforts in artificial intelligence for many years. It is excellent at finding complex structures in huge amounts of data; therefore, it is applicable to many fields of science, business, and government [12]. The relationship among AI, machine learning, and deep learning can be seen in Figure 1 [13,14].

Deep-learning models that are based on a single deep-learning architecture are solo deep-learning models. Models that are produced by connecting two or more deep-learning architectures are hybrid deep-learning (HDL) models. HDL can also be formed by combining one deep-learning model with machine-learning models for classification purposes. This is called the hybridization process [15]. The process of hybridization is often referred to as the process of the fusion of two or more solo deep-learning architectures [16]. Over the course of the last few years, deep learning has grown exponentially in the domain of computer vision and has become a research interest in image recognition issues [17]. The field of computer vision depends a lot on the availability of fine images with labeled datasets by professionals to train, test, and validate algorithms. The limited availability of such datasets is commonly the limiting factor in research and projects [18]. The same is true for the medical field. The availability of professional resources is restricted due to the nonavailability of experts in clinical diagnostics [19].

Much medical practice has been performed with the help of AI to improve the healthcare sector for the past 30 years. Advancements in machine learning and deep learning have also resulted in expanded opportunities for medicine as well [20]. Medical technologies based on AI are growing rapidly and are being used in clinical practice. This covers a wide range of multidisciplinary medical services, from basic clinical practices such as diagnoses to advanced practices such as surgery and remotely treating patients [21]. Medical technologies are aiding healthcare professionals and experts in the identification of some of the deadliest diseases, such as cancer. Early diagnoses, simplification, enhanced treatment, and reduced hospitalization duration are making patient lives easier [22].

Robots are one of the most amazing AI inventions. Just like every aspect of life, robotics is increasingly becoming a part of medicine. AI and robots have vast potential in the healthcare sector. Figure 2 gives a glimpse of the contributions of AI and robotics in the healthcare sector [23]. AI is used by researchers to quickly and efficiently process and respond to data for better treatment outcomes of fatal ailments such as cancer and heart diseases [24].

Cardiovascular diseases (CVDs) or heart diseases are the leading international cause of human death. According to a report conducted in 2019 by the World Health Organization (WHO), approximately 18 million people died that year due to CVDs, representing 32% of total deaths. Among them, 85% was caused by heart failure and stroke [25]. It is difficult for even physician doctors to diagnose these diseases early and correctly. A total of 25% of people die suddenly without any prior symptoms of heart disease. Therefore, it is important to establish a system that can detect heart diseases at an early stage.

Coronary artery disease (CAD) is the most common type of heart disease. Several personal habits, such as smoking, diabetes overuse of alcohol, less or no physical activity at all (obesity), stress, and high blood pressure, can affect the heart and cause diseases [26]. Diagnosing heart disease not only takes a lot of time and effort, but also demands resources. The use of deep learning with image classification techniques can help experts in obtaining valuable input about heart patients and in better diagnosing patients [27].

The objective of this research is the early detection of heart diseases by using a hybrid deep-learning model. The following are the contributions of our proposed work:Early detection of heart disease by utilizing a deep-learning network.Comparison of the proposed work with existing state-of-the-art approaches.Offering a real-time application of the proposed methodology.The proposed method is evaluated using different performance metrics like accuracy, precision, recall, and F1-score.

The rest of the paper structure is organized as follows. Section 2 reports on the work by other researchers on deep learning and heart-disease prediction. Section 3 presents the proposed research methodology and classification technique. Next, the experimental details and results are presented in Section 4. Section 5 comprises the conclusion and future directions.

## 2. Literature Review

Heart-disease prediction is an important ongoing research area. Classification is a significant and crucial decision-making tool in medical science. For the prediction of heart diseases, numerous facts are presented. Scholars are continuing to research this field. Relevant state-of-the-art heart-disease classification approaches are highlighted in the next subsection. Studies indicated that, even in the context of severe monogenic disease such as familial hypercholesterolemia (FH), single-nucleotide polymorphisms (SNPs) can significantly modify the disease phenotype, which may refine cardiovascular disease (CVD) risk prediction in FH patients [28,29]. Both phenotypic and genetic information availability showed promising prediction improvement. Thus, combining phenotypic and genetic information with robust computational models can improve disease prediction in the area under a receiver operating characteristic curve (AUROC) and the area under a precision–recall curve (AUPRC) [30].

Shah et al. [26] proposed a system that utilized some supervised-learning algorithms to predict heart diseases in patients. The basic objective of this study was to predict whether a patient had a chance of contracting heart disease. In this study, the machine-learning UCI repository was used as data, which contains 303 occurrences and 76 attributes. Only 14 attributes were taken for testing—13 were predictors and 1 was a class attribute. After data preprocessing, different data mining algorithms were compared, such as naïve Bayes (NB), decision tree (DT), random forest (RF), and K-nearest neighbor (KNN) to test the data. Among different authors that used similar approaches to this study [31,32,33,34,35,36], the comparison results show that the K-nearest neighbor achieved the highest accuracy of 90.789% for *k* = 7. A similar study focused on coronary heart-disease prediction [37], and used different intelligent computational techniques, such as support vector machine (SVM), NB, logistic regression (LR), deep neural network (DNN), DT, RF, and KNN for prediction and a comparative study. The UCI repository dataset of Statlog [38] and Cleveland [39] was used, which includes both male and female patients. A total of 270 samples were in the dataset that were further divided into 13 attributes and a class distribution attribute. The dataset was divided into training and testing datasets. These datasets were then applied to intelligent computational techniques for model evaluation. The deep neural network technique performed best with the Statlog dataset among all the computational techniques, and attained an accuracy of 98.15%. In the case of the Cleveland dataset, the support vector machine outperformed the others and achieved the highest accuracy of 97.36%.

Another study on medical image classification using AOC-CapsNet to replace the traditional classification method was proposed in [40]. To select low- and high-frequency information, two modules, octave convolution and attention, were used. This study contributed by designing an effective classification framework utilizing AOC-CapsNet for the classification of medical images. The proposed system was tested on seven different datasets. The datasets were divided into training, testing, and validation datasets. AOC-Caps achieved better performance for most of the seven datasets. The proposed methodology, however, did not work well with imbalanced datasets, and the study did not provide a solution to imbalanced data.

Gao proposed a study that worked on imbalanced data [41] using a deep learning approach for medical image classification. Usually, deep-learning techniques require many labeled data classes. There are plenty of medical datasets that do not have balanced data because some of the diseases are not that common. To address the imbalanced-data issue, a deep learning-based approach was proposed that used one-class classification named image complexity-based one-class classification (ICOCC). This was a novel method that used image complexity concepts to learn relevant single-class imaging features. The proposed system was implemented using a convolutional neural network (CNN) framework. The system was tested on four different kinds of medical imaging data. These imbalanced datasets were MRI, FFDM (breast screening datasets), SOKL, and Hep-2. The normal class percentages are 44.4%, 81.8%, 82.3%, and 67.8%, for Hep-2, SOKL, FFDM, and MRI, respectively. The proposed method outperformed existing methods by utilizing the given datasets. ICOCC I produced an AUC of 96.9% with MR. The AUC value for FFDM was 92.4, 70.3 for SOKL, and 94.1% for HEp-2. This showed that the proposed method achieved the best results with the MRI dataset.

In [42], the authors proposed an information-directed synthetic-medical-image adversarial augmentation method for ultrasonography thyroid nodule classification. This study proposed a method to synthesize medical images. An image encoder was designed for radiologists to extract domain information.To synthesize nodule images, this domain knowledge was further utilized to constrain ACGAN. On this basis, thyroid nodule images were classified. Ultrasonic thyroid image data were collected from a local database. A total of 867 patients’ data were used: 591 images with the size of 800×604, and 276 images with the size of 960×720. The generated model thus produced 64 pixel images. The proposed model achieved accuracy of 91.46%. Another deep-learning work on image representation, proposed by [43], was based on electrocardiogram (ECG) images for the classification of arrhythmia. This study proposed a 2D CNN model to classify ECG signals into eight classes. One-dimensional ECG signals were converted into 2D by using short-time Fourier transform. The 2D CNN model was tested on the MIT-BIH arrhythmia dataset that contained 48 patients’ data with a recorded duration of at least 30 minutes. Two experts offered their services to record and annotate the issues. A total of 110,000 descriptions were documented [44]. The two-dimensional CNN model predicted 8 types of arrhythmia with an average accuracy of 99.11% and an average precision of 98.59%.

One issue that researchers face in image classification is the availability of noisy image data. Roy et al. [45] worked on noisy images. Many researchers worked on image classification by using a deep neural network (DNN) due to its layerwise design to extract features from data. DNN also has a similar noise issue. This study aims to develop a DNN-based system for the better classification of noisy images. S.S Roy used five hybrid models, namely, DAE-CNN, CDAE-CNN, DVAE-CNN, DAE-CDAE-CNN, and DVAE-CDAE-CNN, for the classification of noisy images. The MNIST and CIFAR-10 datasets were used for experimentation purposes. The proposed model used the autoencoder-based denoising technique to restructure an image from a noisy image. Then, the convolutional neural network technique of DNN was used for classification. The dataset used for training purposes was affected by 20% of the noise. The DVAE-CDAE-CNN hybrid method performed best among all the classifiers, and it worked better with noisier images. It achieved accuracy of 61.93% with 20% noise in the data, and 53.91% with 50% noisy data. Classifier DVAE- CNN performed better with 20% of noisy data and achieved accuracy of 62.8%, better than that of DVAE-CDAE-CNN, but it achieved 52.64% with 50% of noise.

In [46], Balamurugan, regarding heart-disease prediction, proposed a system that could detect abnormalities in a short amount of time. The dataset was collected from the UCI repository that contained 75 attributes and 303 instances. The preprocessed and normalized data were used for the selection of relevant features. The features were extracted from the medical images using image classification techniques. These features were further clustered by using the adaptive Harris hawk optimization (AHHO) clustering approach. A deep genetic algorithm was then applied to these clustered features for further classification. The proposed system achieved accuracy of 97.3%. This system’s performance was evaluated in the MATLAB/Simulink platform. The precision, sensitivity, and specificity counts for the proposed work were 95.6%, 93.8% and 98.6%, respectively. The prediction of heart disease with DCNN using a method called CardioHelp was proposed by Awais et al. [47]. The purpose of their study was the early detection of heart failure by utilizing a CNN model. The dataset contained 14 attributes for prediction and some cardiac test parameters such as age, gender, cholesterol, and habits. Noisy data were removed by applying different noise-removal techniques such as the data cleansing replacement of anomalies, the calculation of mean values, and normalizers. The CNN model with two classes achieved 97% accuracy; with four classes, it had accuracy of 86.6%.

Fazl-e-Rabbi et al. [48] used multiple classifiers to predict heart diseases. The Cleveland dataset from UCI repository was used that contained 270 records of 76 attributes. This study also used only 13 attributes of the dataset. Three different classifiers were used for the prediction of heart diseases i.e., SVM, artificial neural network, and k-nearest neighbor. The classification accuracy with SVM was 85.18%. The accuracy value with KNN continued to increase while the number of k was increased until k=10. At that point, it reached an accuracy value of 80.74%. Accuracy with the ANN was 73.33%.

In [49], Manimurugan used artificial intelligence and IoMT for the prediction of heart diseases, and proposed a two-stage model. The first stage consisted of collecting data from medical sensors that were connected to patients. A hybrid optimization technique, HLDA-MALO, was used for sensor data classification. The second stage was echocardiogram image classification that used a hybrid R-CNN. This study also used the Cleveland dataset of the UCI repository with 14 attributes. After the application of the model, HLDA-MALO predicted the normal sensor data with 96.85% accuracy, and abnormal sensor data with 98.31% accuracy. The progress of R-CNN was measured with the performance metrics, and it achieved 98.06%, 98.95%, 96.32%, 99.02, and 99.15% for precision, recall, specificity, F score, and accuracy, respectively.

Du et al. [14] worked on the electrocardiographs (ECGs) of heart patients to detect abnormalities. Many researchers had already worked on ECG images, but this study mainly focused on two challenges. Most of the methods are based on digital signal data, whereas most hospital ECG data are stored in image form. The authors proposed a fine-grained multilabel ECG (FM-ECG) framework for the detection of abnormalities in the original ECG images. Detection was conducted in two steps. The first step was the direct detection of abnormalities with the help of a fine-grained classification mechanism. The second step involved taking the ECG label dependencies for further classification. Real-time datasets CECG and DECG were used for experimentation purposes. The F1 score with CECG the dataset was 73.88%; for DECG, it was 86.87%.

Table 1 gives the details of different techniques used by researchers for heart-disease prediction.

As per the results shown in Table 1, most researchers used UCI repository-based datasets and CNN for classification. Image-classification-based prediction is also an important and ongoing research area. CNN models have been used in most works and performed better than the state-of-the-art methods. The accuracy of different works ranged from 53% to 99%. The study in [46] worked on abnormality detection in heart patients and achieved an accuracy of 97.3%. In [49], the authors produced the best accuracy result of 99.15% among all studies. However, this study used sensors to obtain data directly from patients’ bodies. This requires the presence of domain experts while collecting the data. In comparison with other techniques, sensor-based techniques are quite expensive.

## 3. Proposed Methodology

This segment of the paper proposes a framework for the early detection of heart disease. Many techniques have been embraced for predicting heart diseases. In this paper, we use a convolutional neural network (CNN) for the prediction of heart diseases. The data evaluation in systems that are mainly composed of computer-aided diagnosis methods depends on computer-based applications. The transfer of knowledge from a domain expert to a computer algorithm is a manual, tough, and hectic job that mainly depends on the medical expert’s opinion. To efficiently handle this problem, a deep-learning technique, CNN, was used for prediction. Figure 3 shows the architecture of the proposed methodology.

The process starts with the collection of patient data. The data are gathered and enter the preprocessing phase. The missing values are filled or removed using the technique elaborated in the data preprocessing section. Several prepossessing techniques were applied to the dataset for its improvement. The cleaned dataset was further divided into two subdatasets: the training and testing datasets. The training dataset was used for training the model using training data on the CNN algorithm. The testing data were used for evaluating the performance of the proposed model. After model training and testing, the proposed system outputs the result. The proposed system categorizes the patient as healthy or diseased.

### 3.1. Dataset

The dataset of the heart patients used for the proposed methodology is gathered from the University of California (UCI, Irvine C.A) repository, which is publicly available on the Kaggle website (https://www.kaggle.com/datasets/redwankarimsony/heart-disease-data, accessed on 15 June 2022). The dataset collected from Kaggle contained the information of 1050 patients, which included 76 attributes. Out of the 76 attributes, 14 were used for the prediction of heart disease. This is because the other attributes do not affect the disease as much as these attributes. To eliminate the missing and redundant values, the dataset is cleaned and filtered before being used for classification. The dataset was randomly divided into training and testing datasets with 80% and 20% of the samples, respectively. Out of 1025 patient records, 820 records were used for training, and the remaining 205 samples for testing. Training data were used for training the model with the CNN algorithm, and validation data were for validating the performance of the trained model.

### 3.2. Data Preprocessing

Data preprocessing removes any ambiguous data from the collected dataset, resulting in reduced accuracy and prediction rate. Human error could be the reason for the loss of data, which needs to be removed before the training of the model. After the collection of the dataset, it is further preprocessed to remove any missing or duplicate values from the dataset. Unnecessary values are also weeded out, and the dataset is further used for training the model. The deletion of instances is carried out only if the missing value is of the missing-at-random (MAR) type, and those rows are dropped from the dataset where multiple missing entries are present in a single row. For inputting missing values, we employed commonly used mean and median methods for numerical and categorical features, respectively.

### 3.3. Deep Convolutional Neural Network

The architecture of the proposed DCNN model is a feed-forward network with a sequential model in which each layer is connected in a single-input and single-output manner. The heart-disease classification attribute is a binary attribute that is classified as “1” for patients having heart disease, and “0” for patients with the absence of heart disease. The model had 2 convolutional layers followed by 8 dense layers. The 14 selected attributes were joined in the fully connected dense layer. A total of 8 fully connected dense layers were used for building the CNN framework. The first four layers contained 128 neurons, the next contained 64, and the last layer contained 1 neuron. Before the nonlinear transformation, these layers normalize the variables. The exponential linear unit (ELU) was used as an activation function except for in the last layer. The sigmoid function was used as an activation function in the last layer of the CNN model. The Nadam optimizer was used with a learning rate of 0.001. The loss function was set to binary cross-entropy. During the training phase, the number of epochs was set to 100 for better classification. The dropout rate was 3% to avoid overfitting by the model.

### 3.4. Sigmoid Function

The sigmoid function is a mathematical function that has an S-shaped curve and can map any numerical value into a small range of numbers i.e., between 0 and 1. The logistic function of the sigmoid function is mostly used in neural networks, so it is also referred to as the logistic sigmoid function [46]. It was used as the last layer in our CNN model to transform the outcome of our proposed model into a probability score. The formula for a sigmoid function [50] and its graphical representation are described in Figure 4.

### 3.5. Nadam Optimization Algorithm

Nadam is an extension of the Adam optimizer that smooths out the noisy objective function and improves convergence. The Nadam optimization algorithm was used alongside our proposed CNN algorithm to reduce the overall loss and improve the accuracy of the model. It uses the same learning rate alpha for updating weights and is a little faster in training time than Adam.

## 4. Results and Discussion

The dataset from the UCI repository was obtained from the Kaggle dataset inventory and contained a number of attributes, listed in Table 2, with a possible description combined by medical experts. Out of the 76 attributes, only 14 were selected with a regression analysis method called Lasso. Before starting the experiment, the dataset needed to be prepared. Some irregularities and noise were found in the dataset that needed to be removed before we used these data for experimentation. The preprocessing step involves removing anomalies from the dataset by replacing missing values with mean values. The encoding process of preprocessing converted the categorical data into numeric data. Once the dataset had been prepared, it was split into training and testing data. Training data were used for training the classifier generated by using the proposed model. The model classified the data into two binary classes: “1” for patients suffering from HD, and “0” for patients having no heart disease. The computed prediction accuracy by the model was 91.71%.

### 4.1. Experimental Setup

The experimental setup of the proposed model was established to discover and perceive the performance of the convolutional neural network (CNN) with the dataset available at the UCI repository. Table 2 shows the attributes used in the dataset and its description.

### 4.2. Performance Metrics

The performance of the proposed model could be evaluated by using some performance metrics. In deep learning, there are various standards for evaluating the performance of a system. Some performance metrics i.e., accuracy, precision, recall, and F1 score, are discussed in the subsequent sections.

#### 4.2.1. Accuracy

One way to gauge how frequently a machine-learning algorithm correctly classifies a data point is to evaluate the algorithm’s accuracy, which is the ratio between number of correct predictions over the total prediction, as given in Equation (Equation 1).
(1)$$Precision=\frac{TP+TN}{TP+TN+FN+FP}$$

#### 4.2.2. Precision

Precision (also known as positive predictive value) is the measure of the number of correct predictions to the total number of inputs. The precision can be calculated by using Equation (Equation 2).
(2)$$Precision=\frac{TP}{TP+FP}$$

#### 4.2.3. Recall

Recall (also known as sensitivity) represents the ratio of correct class predictions to the total number of inputs of that class. It is a measure to determine the completeness of the classifier. Recall can be calculated with the help of Equation (Equation 3).
(3)$$Recall=\frac{TP}{TP+FN}$$

#### 4.2.4. F1-Score

It can sometimes be difficult to decide whether high precision or low recall is better, or vice versa when comparing different models. The combination of precision and recall is called the F1 score. It can be measured by using Equation (Equation 4).
(4)$$F1-Score=2\times \frac{\left(Precision\times Recall\right)}{Precision+Recall}$$

In Equation (Equation 1), *TP*, *TN*, *FP*, and *FN* denote true positive, true negative, false positive, and false negative, respectively.

### 4.3. Classification Using CNN

Once the dataset had been prepared, it was used for further execution of the model. The proposed model was implemented and trained on Google Collaboratory (Google Colab) using the CNN algorithm. The whole model was implemented in Python and TensorFlow. CNN was used as the backbone architecture for the model. We also added one multihead self-attention mechanism with 8 heads. We used the Nadam optimization algorithm [51] along with CNN for classification. The initial learning rate was set to 0.001. The batch size was set to 32, and the epochs were set to 100. The model yielded an accuracy of 90.12% on the training data, and 91.71% on validation data after its successful training. The given confusion matrix in Figure 5 was used to define the performance of the proposed system.

Table 3 depicts the performance of the proposed model in terms of precision, recall, and F1 score. The overall accuracy of the system was 91.71%, as shown in Table 3. Precision, recall, and F1 score for the proposed model were 88.88%, 82.75%, and 85.70%, respectively.

To better determine and evaluate the proposed system, Table 4 illustrates the performance comparison between state-of-the-art approaches and the proposed system relating to accuracy outcomes, which shows that the proposed technique outperformed the other approaches.

Another way to evaluate the proposed system is by evaluating the value of training loss and validation loss. To see how well the proposed system is fitting the unseen data; we plot the training and validation loss on a graph. Its visual representation can be seen in Figure 6. It depicts the training loss to be 24.02 and the validation loss to be 17.39%.

Figure 7 shows the training and validation accuracy represented in the form of a graph. Training accuracy was 90.12%, and accuracy after data validation was 91.91%.

The performance of the classification algorithms is intrinsically linked with the area under the curve (AUC), i.e., the larger the value of the AUC is, the better the performance of the classification algorithm. The proposed model’s strength and robustness were evaluated using the ROC curve. The receiver operator characteristics curve of the proposed model is shown in Figure 8.

## 5. Conclusions

The prediction of heart disease at an early stage can prevent many mishaps. The use of an efficient algorithm can help physicians in detecting the possible presence of heart disease before it manifests. This research focused on using a state-of-the-art UCI repository for the early detection of heart diseases. Initially, the dataset was collected from the UCI repository with 76 instances, and14 of which were used for the prediction. Before training the data, they were preprocessed. The preprocessed data were used with the CNN algorithm for predicting heart diseases on Google Collab. The proposed system was evaluated regarding the performance metrics of accuracy, precision, recall, and F1 score, and in which it achieved 91.71%, 88.88%, 82.75%, and 85.70% respectively.

## Figures and Tables

**Figure 1 biomedicines-10-02796-f001:**
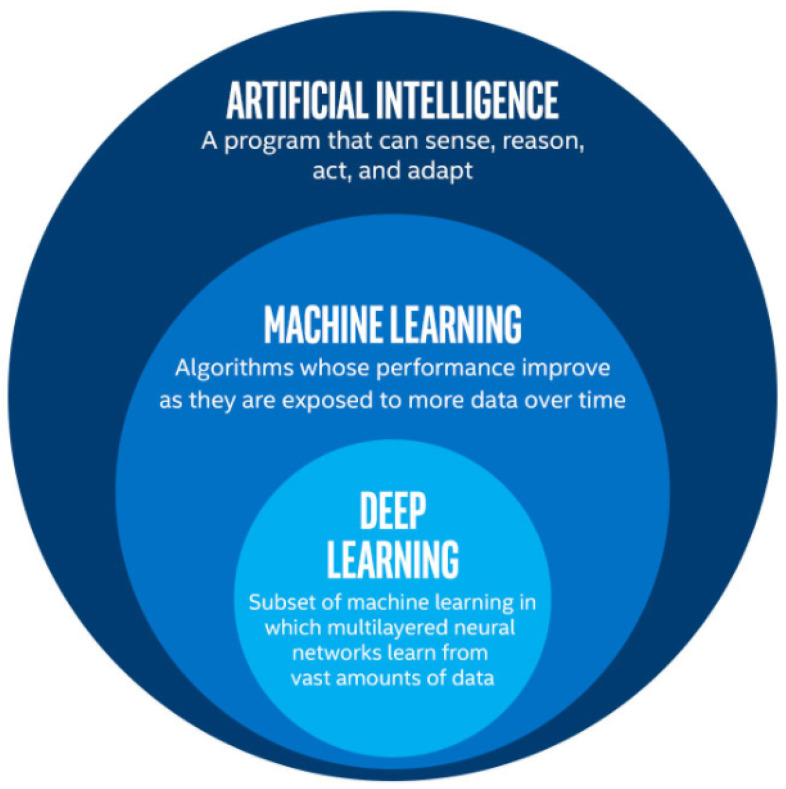
Relationship among artificial intelligence, machine learning, and deep learning.

**Figure 2 biomedicines-10-02796-f002:**
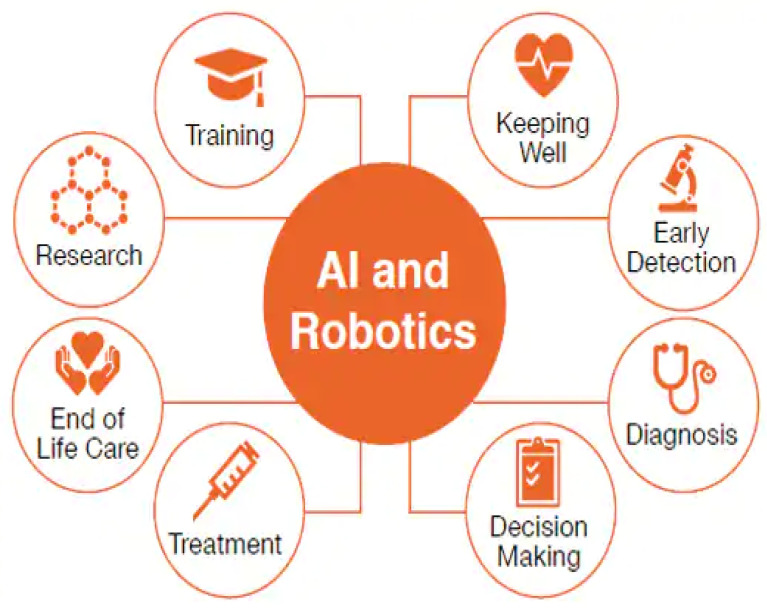
Artificial intelligence and robotics in the healthcare sector.

**Figure 3 biomedicines-10-02796-f003:**
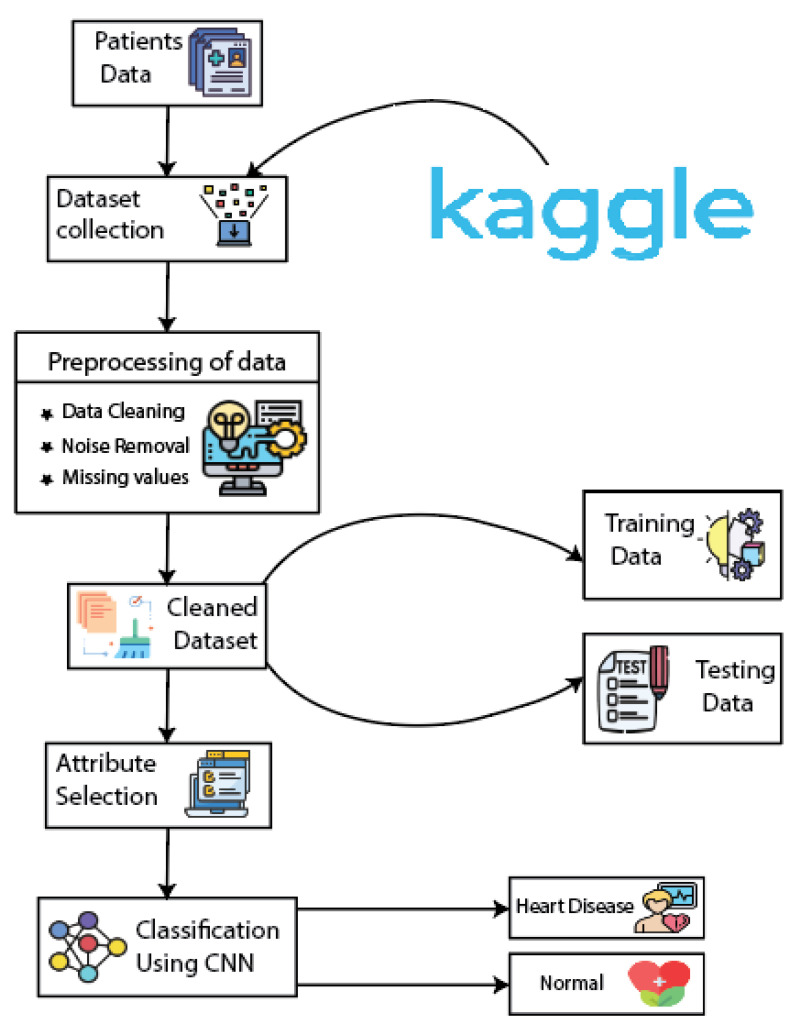
Proposed methodology for heart-disease prediction.

**Figure 4 biomedicines-10-02796-f004:**
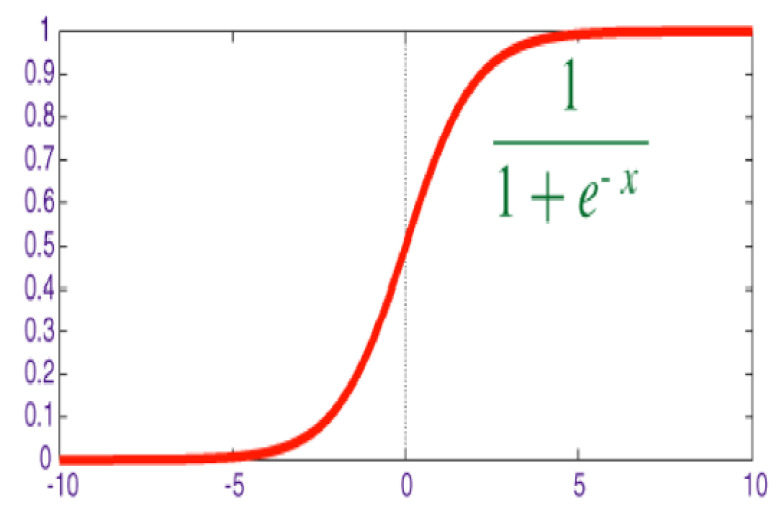
Sigmoid function.

**Figure 5 biomedicines-10-02796-f005:**
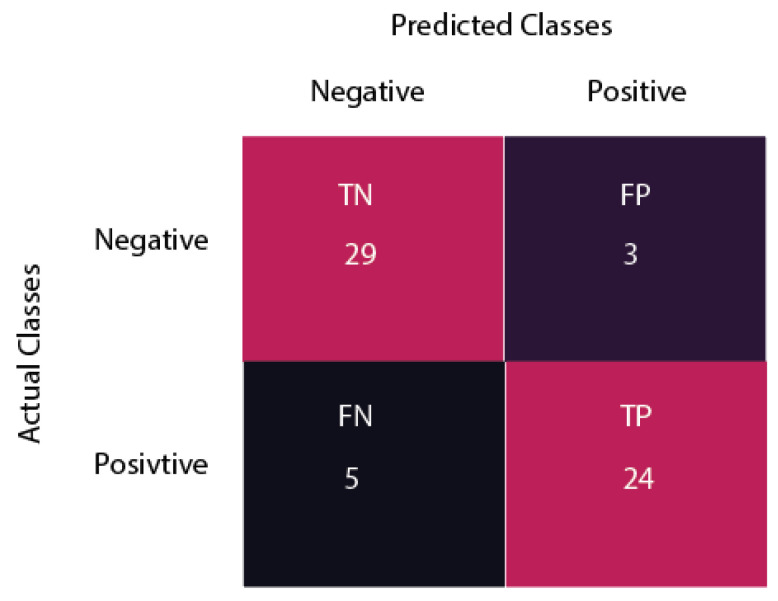
Confusion matrix of the proposed system.

**Figure 6 biomedicines-10-02796-f006:**
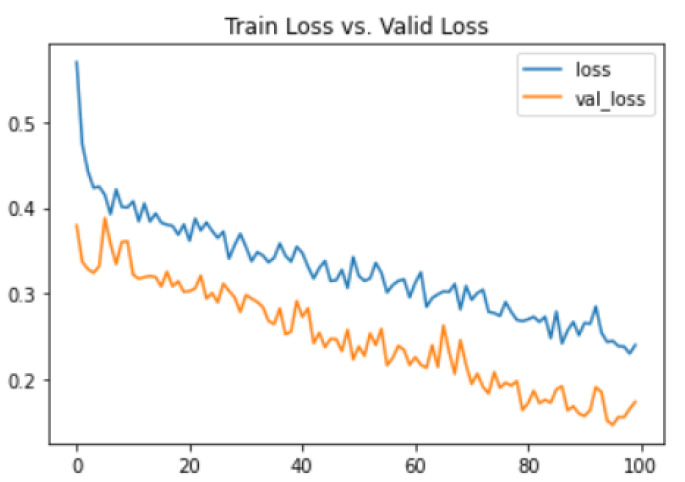
Graphical representation of training and validation loss.

**Figure 7 biomedicines-10-02796-f007:**
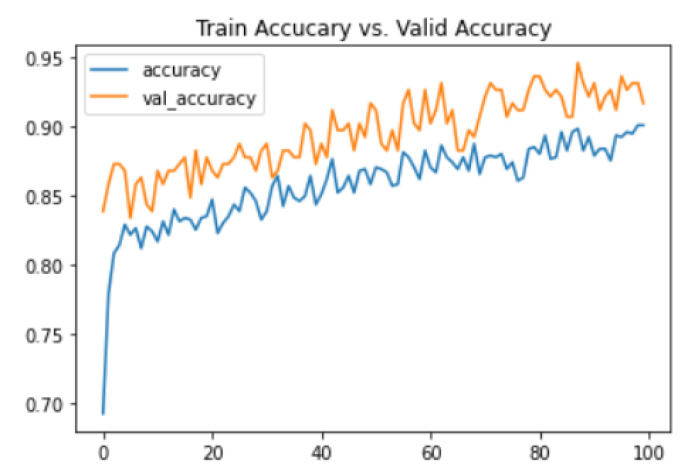
Graphical representation of training and validation Accuracy.

**Figure 8 biomedicines-10-02796-f008:**
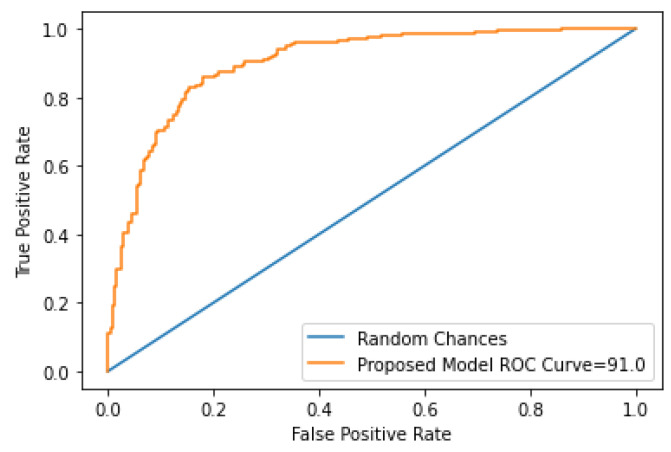
ROC curve of the proposed model.

**Table 1 biomedicines-10-02796-t001:** A relative study of different literature reviews.

Ref	Objective	Techniques	Accuracy %	Precision %	Recall %	F1 Score %
[26]	The early detection of cardiovascular disease in patients.	NB, DT, DF, and K-NN classifiers	KNN:90.7, DT: 80.2, RF: 84.2, NB: 88.15	N/A	N/A	N/A
[34]	A comparative study of intelligent computational techniques	SVM, NB LR, DNN, DT, RF, and K-NN.	SVM:97.41, NB: 91.38, LR: 96.29, DNN: 98.15, DT: 96.42, RF: 90.46, KNN: 96.42	N/A	N/A	N/A
[37]	Medical image classification	AOC-CapsNet	93.1	92	90.3	91.9
[38]	Handling imbalanced medical images	CNN framework	N/A	N/A	N/A	N/A
[39]	Ultrasonography thyroid nodule image synthesis	KACGAN-based model	91.46	N/A	N/A	N/A
[40]	Classification of arrhythmia	2-D CNN	99.11%	98.58	N/A	98
[42]	Classification of noisy images	Five hybrid CNN models	DVAE- CNN: 62.8, DVAE-CDAE-CNN: 53.91	N/A	N/A	N/A
[43]	Heart-disease prediction	AHHO and deep genetic algorithm	97.3	95.6	N/A	N/A
[44]	Heart-disease prediction	CNN	97	N/A	N/A	N/A
[45]	Heart-disease prediction	ANN, SVM, and KNN	SVM: 85.18, KNN: 80.74, ANN: 73.33	N/A	N/A	N/A
[46]	AI and image-classification-based heart-disease prediction	HLDA-MALO and hybrid R-CNN with SE-ResNet-101 model	99.15	98.06	99.15	99.02
[47]	Detection of abnormalities in ECG images.	FM-ECG framework	CECG: N/A, DECG: N/A	79.23, 90.42	69.10, 83.59	73.88, 86.87

**Table 2 biomedicines-10-02796-t002:** Description of the attributes used in the dataset.

Sr No.	Attributes	Representation	Description	Type
1	Age	age	Age in years	Integer
2	Gender	sex	Male and female	Binary(1 for male and 0 for female)
3	Chest pain	cp	Four types of chest pain	Categorical
4	Cholesterol level	Chol	Measure of cholesterol in mg/dl	Integer
5	Resting blood pressure	trestbps	Blood pressure when the body is in a state of rest	Integer
6	Fasting blood sugar	fbs	Blood sugar level while fasting	Binary (1 for true and 0 for false)
7	MaxHR	thalach	Maximal heart rate	Integer
8	Rest ECG	restecg	Resting electrocardiograph	categorical
9	Exercise-induced angina	exang	Exercise-induced angina	Binary (1 for yes and 0 for no)
10	Old peak	oldpeak	ST depression brought by exercise comparative to rest	Continuous
11	Slope	slope	Slope of exercise peak	Discrete
12	Vessels	ca	No. of major vessels	Continuous
13	Thalassemia	thal	Normal, fixed, and reversible defects	discrete
14	Heart disease	target	Predicted attribute	Binary

**Table 3 biomedicines-10-02796-t003:** Performance metrics of the proposed model.

Performance Metrics	Accuracy (%)
Accuracy	91.71
Precision	88.88
Recall	82.75
F1 score	85.70

**Table 4 biomedicines-10-02796-t004:** Comparison between the performance of the proposed method and that of existing methods.

Author (Technique)	Accuracy (%)
Fazle Rabbi et al. [46] (artificial neural network)	73.3
Qrenawi, Mohammed et al. [51] (Rmonto ontology-driven data mining)	90.0
Kaanchan More et al. [50] (risk-factor-based approach)	86.7
Weize Xu et al. [52] (random forest and Adaboost classifier)	78.4
**Proposed approach**	91.7

## Data Availability

Not applicable.

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
