# Peer review of "A Deep Convolutional Neural Network for the Early Detection of Heart Disease"

_biomedicines, 2022, doi:10.3390/biomedicines10112796_

Round 1
Reviewer 1 Report
The article ”A Deep Convolutional Neural Network for the Early Detection of Heart Disease” by Sadia Arooj and colleagues tackles an interesting clinical challenge and presents promising results. However, the presentation of the methods is, on occasions, scarce and must be supplemented by a decision curve analysis in order to become clinically more convincing. The shown tables and figures are pedagogic. Please find detailed comments below.
Abstract. Information of the used dataset are missing. This information follows first in lines 274-6. Please add also here.
Line 13. Please add that your approach outperformed four earlier proposals in terms of accuracy.
L.31. A.I. should probably read AI
L.33-7. More data does definitely NOT automatically implicate better data. It is acknowledged that an algorithm-based solution might be able to find patterns that the human observer would not; however, good input data is still required especially in big data situations where the data more often than not is the result of a form of convenience sampling, meaning NOT sampled for the purpose at hand. Data-wise “garbage in, garbage out” always applies.
L.73. Delete superfluous comma.
L.75. Replace medicine by health care.
L.79-80. The literature reference #50 in the legend of Fig. 2 should be #23 in order to keep the chronological order of cited references.
L.90-1. Do the authors refer to symptomatic coronary heart disease and/or asymptomatic coronary heart disease? Especially the latter is not only a matter of time, effort, and resources, but clinically just very difficult indeed. If not impossible at times!
L.130-3, 144, 148, 156-7, 208, 213-4. The section on the literature review is much appreciated, thank you! However, please add information on what kind and size of data the former works were based on; which patients were studied (asymptomatic, symptomatic)? How many subjects were used? Was the material used for training and validation? If so, how was the material divided into strata for different purposes (see, e.g. l.133, 144)? Altman and colleagues (https://doi.org/10.1136/bmj.b605) distinguished, for instance, between internal, temporal, and external validation. Information of how former works were validated will more transparently indicate the limitations of the work that you compare your approach to.
L.172. Replace 91.46Another by 91.46%. Another
L.213. Likewise: replace 86.6Fazl-e-Rabbi by 86.6%. Fazl-e-Rabbi
Table 1. Please add information of patient population and sample sizes for each study shown.
Fig. 3. Please replace GOOGLE by GOOGLE (Kaggle) to be more precise.
L.263. Missing values were filled and removed – how were missing values imputed, when (and why) were data removed? Just reading l.263 leaves the impression of making the data fit for purpose, possible involving untoward making the data easier to get separated into patients with and without coronary heart disease later, meaning deleting the difficult cases from the grey zone.
L.265: how??
L.269: what is the benchmark probability for declaring a patient as diseased? 50% disease probability?
L.270. Replace : by .
L.277-8. how?? More elaborate descriptions can (and should) be added as Supplemental Material.
L.279-80. As Altman and colleagues (see link above) pointed out will a random split just mean that you get two datasets that are quite alike. Please apply at least temporal validation to secure a basic level of validation, thanks. Splitting the data 80-20 (training-validation) is fine.
L.284-90. See comment to l.277-8.
L.296: This information – that a probability score of your approach becomes available – is a crucial information. This means that decision curve analysis is possible and must supplement your otherwise nice analysis strategy. Please consult respective work by Andrew Vickers (Mayo Clinic), for instance, doi: 10.1007/s00330-022-08685-8 ; doi: 10.1016/j.spinee.2021.02.024 ; doi: 10.1186/s41512-019-0064-7 ; doi: 10.1200/JCO.2016.69.1576 ; doi: 10.1053/j.seminoncol.2009.12.004 ; doi: 10.1198/000313008X37030 ; doi: 10.1186/1472-6947-8-53 ; doi: 10.1002/sim.3087 ; doi: 10.1177/0272989X06295361
L.310: please add decision curve analysis as another metric. This will help the reader see the performance of your approach across the range of threshold probabilities (for which I assume you simply used 50% in the classification of heart disease patients vs. healthy subjects, see above).
L.313: Replace : by .
Table 2. Check and revise, for instance, Sr(?) No. 14, representation equals target, description is The
L.314. Replace Precision by Precision (also known as positive predictive value)
L.315. Replace Recall by Recall (also known as sensitivity)
L.327-8. Does the accuracy of 90.12% relate to a cut-off of 50% for disease probability (see also l.269)?
Table 4. Replace 73.33 by 73.3 to harmonize formatting in the table.
L.331. Replace table 2 by Table 2
L.328. Replace figure 5 by Figure 5.
Figure 5. Please add a brief explanation in the legend of what the confusion matrix represents. What are the 29+3+5+24 = 61 elements shown in the table? Please explain all 4 abbreviations TN, FP, FN, and TP (as also partly done in l.317-8 where TN was not used yet).
L.410. Ref. #23 extends beyond the readable part of the page.
Author Response
Thank you for the opportunity to revise our manuscript. We appreciate the careful review, constructive suggestions, and insightful comments. In the revised manuscript we have made almost all of the changes the reviewer has suggested. We believe that the manuscript is substantially improved after making the suggested edits. Changes made in the revised manuscript are shown in red color.

Reviewer 2 Report
Thank you for the opportunity to review this paper. In this paper, the authors aimed to use a deep learning approach using image classification for heart disease detection. This study is confusing, and the presentation's quality is vague. It can not be considered for publication in this form.
I have lots of comments, but I am pointing out several significant comments here:
Abstract:
1. Background is too much (line 1-7), which are not related to the study. Line 9: "A deep convolutional neural network (DCNN) ........" I don't know why it comes after the study's objective.
Line 10-11: "The proposed method is concerned with....." What do you mean by that?
Line 12: "For gathering information from the dataset, data mining techniques are applied to........" data mining is broad terms? you have only used the DCNN algorithm?
No conclusion in the abstract section.
Introduction:
Too much unnecessary information, like in figure 1 and figure 2.
Line 98-99: "Validation of the outcome of the proposed work by using different performance metrics like precision, recall, and F1-score". The validation process can be done using different datasets or a small portion of data from the same database. precision, recall, and F1-score are just metrics to measure your findings.
Methods:
I don't think figure 3 is right.
Section 3.1: Lack of information about data, no link for data. Did not describe how you have boiled down the total number of attributes to 14. Which method have you used to feature selection?
3.2: How have you handled missing data? what was the process of handling missing data? duplicate value? How have you removed duplicate values?
lack of information on how authors have developed the CNN model?
No discussion.
Author Response

(The authors gave the same response as above.)

Round 2
Reviewer 1 Report
Dear authors. Thank you for providing a thorough update of your work. The manuscript did clearly improve, much appreciated. Regrettably, you decided against the conduct of a decision curve analysis which I would have loved to see. Sorry, I forgot to mention the respective and very helpful homepage for it: www.decisioncurveanalysis.org. However, the ROC curve analysis (Fig. 8) palliates at bit, at least. Congratulations on your work and best wishes.
Reviewer 2 Report
Thanks for your revised version.